



# Comparison of middle- and low-latitude sodium layer from a ground-based lidar network, the Odin satellite, and WACCM-Na model

Bingkun Yu[1,2], Xianghui Xue[1,3,4,5,6], Christopher J. Scott[2], Mingjiao Jia[7], Wuhu Feng[8,9], John M. C. Plane[8], Daniel R. Marsh[8,10], Jonas Hedin[11], Jörg Gumbel[11], and Xiankang Dou[1,12]

[1]CAS Key Laboratory of Geospace Environment, Department of Geophysics and Planetary Sciences, University of Science and Technology of China, Hefei, China
[2]Department of Meteorology, University of Reading, Berkshire, UK
[3]CAS Center for Excellence in Comparative Planetology, Hefei, China
[4]Anhui Mengcheng Geophysics National Observation and Research Station, University of Science and Technology of China, Hefei, China
[5]Hefei National Laboratory for the Physical Sciences at the Microscale, University of Science and Technology of China, Hefei, China
[6]Frontiers Science Center for Planetary Exploration and Emerging Technologies, University of Science and Technology of China, Hefei, China
[7]Shandong Guoyao Quantum Lidar Co., Ltd., Jinan, Shandong, China
[8]School of Chemistry, University of Leeds, Leeds, UK
[9]National Center for Atmospheric Science, University of Leeds, Leeds, UK
[10]National Center for Atmospheric Research, Boulder, CO, USA
[11]Department of Meteorology, Stockholm University, Stockholm, Sweden
[12]Electronic Information School, Wuhan University, Wuhan, China

**Correspondence:** Bingkun Yu (bkyu@ustc.edu.cn); Xianghui Xue (xuexh@ustc.edu.cn)

**Abstract.** The ground-based measurements obtained from a lidar network and the six-year OSIRIS limb-scanning radiance measurements made by the Odin satellite are used to study the climatology of the middle- and low-latitude sodium (Na) layer. Up to January 2021, four Na resonance fluorescence lidars at Beijing (40.2°N, 116.2°E), Hefei (31.8°N, 117.3°E), Wuhan (30.5°N, 114.4°E), and Haikou (19.5°N, 109.1°E) collected vertical profiles of Na density for a total of 2,136 nights (19,587 h). These large datasets provide routine long-term measurements of the Na layer with exceptionally high temporal and vertical resolution. The lidar measurements are particularly useful for filling in OSIRIS data gaps since the OSIRIS measurements were not made during the dark winter months because they utilise the solar-pumped resonance fluorescence from Na atoms. The observations of Na layers from the ground-based lidars and the satellite are comprehensively compared with a global model of meteoric Na in the atmosphere (WACCM-Na). The lidars present a unique test of OSIRIS and WACCM, because they cover the latitude range along 120°E longitude in an unusual geographic location with significant gravity wave generation. In general, good agreement is found between lidar observations, satellite measurements, and WACCM simulations. Whereas the Na number density from OSIRIS is slightly larger than that from the Na lidars at the four stations within one standard deviation of the OSIRIS monthly average, particularly in autumn and early winter arising from significant uncertainties in Na density retrieved from much less satellite radiance measurements. WACCM underestimates the seasonal variability of the Na layer





observed at the lower latitude lidar stations (Wuhan and Haikou). This discrepancy suggests the seasonal variability of vertical constituent transport modeled in WACCM is underestimated because much of the gravity wave spectrum is not captured in the model.

## 1 Introduction

The layers of neutral metal atoms exist in the Earth's mesosphere and lower thermosphere (MLT) at altitudes between 75–
110 km (Plane, 2003; Plane et al., 2015, 2018). The MLT region is of particular importance because it marks the transition between the lower neutral and upper ionised atmospheres. Metal layers in the MLT can provide a unique way to study the chemical and dynamical processes in this region. The metals ablating from meteoroids entering the atmosphere include Na, Fe, Mg, Al, Ni, Ca, and K (Plane et al., 2015). The first quantitative measurements of metal atoms were made with ground-based photometers in the 1950s by the observations of resonance scattering of sunlight (Hunten, 1967). Na has a suitably large
resonant scattering cross-section and a high column abundance as one of the primary meteoric species in the MLT. As a result, the Na atom layer was the first to be discovered and is one of the most researched metal layers (Bowman et al., 1969; Sandford and Gibson, 1970).

Over the last two decades, the space-borne limb-scanning spectrometers have provided a near-global measurement of Na layers. The first global geographical distribution of Na layers was produced by the Global Ozone Monitoring by Occultation of
Stars (GOMOS) spectrometers onboard the Envisat satellite (Fussen et al., 2004), followed by a more detailed study based on seven years of GOMOS observations (Fussen et al., 2010). The Optical Spectrograph and Infra-Red Imager System (OSIRIS) spectrometer onboard the Odin satellite measures the Na concentration profiles from the limb-scanning observations of Na D-lines at 589 nm in the dayglow (Gumbel et al., 2007; Fan et al., 2007a, b; Plane, 2010; Hedin and Gumbel, 2011; Langowski et al., 2017). The Na profiles retrieved from Odin/OSIRIS typically have an altitude resolution of 2 km and an accuracy of 20%
in the metal density at the layer peak (Gumbel et al., 2007). Koch et al. (2022) presented the comparison of Na measurements between OSIRIS/Odin, GOMOS/Envisat, and SCIAMACHY/Envisat. The Na concentration profiles from three space-borne instruments agree well.

Since the first Na lidar measurements in England in the 1960s (Bowman et al., 1969), the light detection and ranging (lidar) technique has permitted long-term routine monitoring of the Na layer (She et al., 2000; Vishnu Prasanth et al., 2009). Many
studies of the diurnal and seasonal variations in the Na layer (Gardner et al., 2005; Yuan et al., 2012; Plane et al., 2015; Li et al., 2018) and the dynamical and chemical coupling between ionized and neutral metals (Yuan et al., 2014b; Yu et al., 2019a; Xia et al., 2020) have resulted from the substantial growth in ground-based Na lidar stations making Na measurements in recent decades. Sporadic sodium layers (SSLs), characterised by a sudden increase of Na atoms in a narrow altitude range have been of particular interest. The SSL was first reported by Clemesha et al. (1978) from Na lidar observations, and such an extremely
narrow and sharp increase in Na atom density could not be produced by neutral density perturbations. Therefore the sporadic E (Es) layers, thin layers of concentrated metallic plasma in the lower E region of the ionosphere, are proposed to be a source of sodium in the mesosphere (Plane, 2012). Studies show that the SSLs are closely related to the changes in $Na^+$ metallic ions





within Es layers (Yuan et al., 2014b; Qiu et al., 2018; Xia et al., 2021). From a laboratory kinetic study, Cox and Plane (1998) demonstrated that the sporadic metal layer is produced when the Es layer descends below 100 km under the influence of tides.

Further observations have confirmed the strong coupling between the SSLs and Es layers (Cox and Plane, 1998; Dou et al., 2010; Williams et al., 2007; Dou et al., 2013; Yuan et al., 2013; Qiu et al., 2016, 2021b). The simulation results show that the variation in the climatology of Na density in summer at mid-latitudes is modulated by the convergence of metallic ions with a descending Es layer through the prevailing descent of tidal wind shear (Cai et al., 2017). Na lidars can provide continuous local measurements of the Na layer with exceptionally high temporal and vertical resolution (usually 3 minutes and 0.1 kilometres),

and only provide point measurements, which is a severe constraint. As a result, it is not possible to obtain a global distribution of the Na layer using ground-based lidars.

Many modelling studies have focused on simulating the seasonal and short-term variability of metallic ions and atoms (Feng et al., 2013; Marsh et al., 2013; Cai et al., 2019a, b). Global climate models generally focus on large-scale advection and eddy diffusion transport, but they do not resolve the omnipresent small-scale waves and turbulence fluctuations (Marsh et al., 2007).

However, dynamical and chemical constituent transport in the MLT is significantly impacted by dissipating gravity waves (Gardner and Liu, 2016). It has been shown that the meteoric influx employed in the Whole Atmosphere Community Climate Model (WACCM) is much smaller, by a factor of approximately 5, than the actual vertical flux from Na lidar measurements (Gardner and Liu, 2014), which suggests that the vertical transport of atmospheric constituents (e.g. Na and Fe) as modeled in WACCM is too slow (Gardner and Liu, 2016).

A ground-based network of four Na lidars has been routinely operated in the Chinese Meridian Project (CMP) over the last decade (Wang, 2010). The lidars range from low- to mid-latitudes roughly along 120°E longitude in China. These lidars are in an unusual geographic location with significant gravity wave generation (Hocke et al., 2016; Zeng et al., 2017). By the year 2021, a total of 2,136 nights (19,587 h) of vertical profiles of Na density were obtained. The lidar observations provide an important test of Odin satellite measurements and WACCM simulations, allowing us to investigate the impact of waves and

turbulence fluctuations on the WACCM predictions of Na density. It should be noted that the original ground-truthing of Odin measurements involved Na lidar measurements at Fort Collins, Colorado which is at a latitude of 41°N (Gumbel et al., 2007).

The present paper is therefore a study of the latitudinal and seasonal variations of the Na layer at mid- and low-latitudes, using a combination of the Odin satellite measurements, the long-term ground-based measurements obtained from the lidar network, and the WACCM-Na model. The long-term Na lidar data are compared to the sun-synchronous satellite measurements

at descending and ascending nodes and simultaneous WACCM simulations. Sect. 2 describes the instruments and datasets. In Sect. 3, the global climatology of the Na layer from OSIRIS spectrometer with a high spatial resolution is presented for comparison with the WACCM-Na simulations, then validated by the ground-based lidar measurements, with a focus on the geographical distribution of the layer, and the seasonal, monthly, and diurnal variations in Na density. We also examine the link between Es layers and SSLs. Sect. 4 summarizes the conclusions of this study.



## 2   Data and Method


The CMP deployed a chain of four Na resonance fluorescence lidars along 120°E longitude at Beijing (40.2°N, 116.2°E) (Gong et al., 2013), Hefei (31.8°N, 117.3°E) (Dou et al., 2009), Wuhan (30.5°N, 114.4°E) (Yi et al., 2013), and Haikou (19.5°N, 109.1°E) (Jiao et al., 2017). The transmitter of the lidar system is a frequency-stabilized dye laser pumped by a Nd:YAG laser, tuned onto the Na D2-resonant absorption line at 589.6 nm to excite resonant fluorescence from Na atoms between 75–110 km

altitude. A telescope collects the backscattered photons, which are then recorded by a photomultiplier tube using an interference filter centred at 589 nm. The CMP lidars can only provide night-time measurements since they do not operate in daytime. Up to January 2021, the four Na lidars have provided the long-term routine measurements for a total of 2,136 nights (19,587 hours). Figure 1 shows the monthly variations in the number of observations measured by the four lidars. Because clear weather is required for Na lidar measurements, and there is a regular presence of convective weather and thunderstorms during summer,

there is a higher measurement coverage during winter. Therefore, the number of valid Na observations from lidars exhibits a seasonal variation. Table 1 summarises the Na lidar data used in this study as well as the lidars' primary parameters.

Odin is a limb-scanning satellite co-funded by Sweden, Canada, France, and Finland (Murtagh et al., 2002). On February 20, 2001, the satellite was launched from Svobodny, Russia. It is in a sun-synchronous orbit at approximately 600 km, with a descending node at 06:00 local time (LT) and an ascending node at 18:00 LT. The satellite conducts limb scans from 10 to

110 km altitude. The OSIRIS spectrometer is one of the two main instruments onboard (Llewellyn et al., 2004). The instrument measures radiance from the limb at wavelengths between 280 and 800 nm. The profiles of mesospheric Na number density have been retrieved from the limb-scanning measurements of the Na D dayglow at 589 nm with an altitude resolution of 2 km (Gumbel et al., 2007; Hedin and Gumbel, 2011), using an optimal estimation method (Rodgers, 2000). Figure 2 shows the time-latitude distribution of the number of the Na observations from OSIRIS during 2004–2009, using a 5 day × 5° latitude grid as a

resolution. The satellite orbits do not fully cover latitudes greater than 85°. Furthermore, mesospheric dayglow measurements cannot be made at mid- to high-latitudes in the winter hemisphere, due to the lack of daylight when the solar zenith angle is larger than 92°. Therefore, the ground-based Na lidars, in addition to OSIRIS limb-scanning radiance measurements, provide an important measurement of local Na layers, notably in filling in data gaps during the winter.

A global atmospheric model of the meteoric metals Fe, Na, K, Si, Ca, Al, Mg and Ni (e.g., Feng et al., 2013; Marsh

et al., 2013; Plane et al., 2014; Langowski et al., 2015; Plane et al., 2016, 2018; Daly et al., 2020; Plane et al., 2021) has been developed, in order to advance understanding the meteor astronomy, atmospheric chemistry, and transport processes that control the different metal layers in the MLT (Feng et al., 2015; Plane et al., 2015; Wu et al., 2019). The model uses a seasonally varying meteoric injection rate of these metals at different latitudes and altitudes. In the present paper, we use a version of WACCM-Na (Marsh et al., 2013) nudged with NASA's Modern-Era Retrospective Analysis for Research and

Applications (MERRA2) (Molod et al., 2015). Here we run the model with a very high vertical resolution (144 vertical levels) of ∼ 500 m in the MLT (e.g., Yuan et al., 2019); and the horizontal resolution is 1.9° in latitude by 2.5° in longitude. The model run covers the time period from January 2004 to the end of 2009. Here we sampled the model output every 30 mins for





the locations of the CMP lidar stations. The simulated Na number density is then used to compare with the observations from the Odin satellite and the Na lidars.

There is strong evidence for close coupling between the metal layers and ionospheric Es layers (Xue et al., 2017; Qiu et al., 2018). Es layers are thin-layers of highly concentrated plasma composed of metallic ions and electrons, that occur between 90–130 km altitude. At mid-latitudes they are caused by the vertical convergence of ions at the null points of wind shears (Whitehead, 1960; Davis and Johnson, 2005; Chu et al., 2014; Yu et al., 2021c). Strong plasma irregularities have a considerable impact on the amplitude and phase of radio occultation (RO) signals from the global navigation satellite system

(GNSS) (Yue et al., 2015, 2016). Therefore, the ionospheric effects on GNSS RO signals can be used to investigate the global ionosphere's morphology. The GNSS RO data from the Constellation Observing System for Meteorology, Ionosphere and Climate (COSMIC) satellites during 2006–2014, have been used to study Es layers (Yu et al., 2020). The maximum S4 index occurring between 90–130 km can be used as a proxy for the intensity of an Es layer (Yu et al., 2019b, 2021b; Qiu et al., 2021a). In the present study we also looks into the climatology and seasonal variability of metallic ions within the Es layers,

as well as the link between Es layers, the Na layer, and the presence of SSLs at low- and mid-latitudes.

## 3   Results and Discussion

### 3.1   Global map and seasonal variation of Na layers

Figure 3 shows the time-latitude distribution of the Na column number density from the OSIRIS limb-scanning radiance measurements between 2004 and 2009, with a resolution of 5 day $\times$ 5° latitude grid. The Na column density is integrated

from 76 to 106 km altitude. Grey-shaded areas represent regions where OSIRIS limb-scanning radiance measurements were not made in winter. The time-latitude distribution of the Na column number density from OSIRIS shows a distinct annual cycle of the Na column density at mid- and high-latitudes, consistent with previous ground-based and satellite measurements (Plane et al., 1999; Fan et al., 2007b; Li et al., 2018). At latitudes between 30° and 90°, the annual variation in Na column density becomes very significant. The maximum Na column density during winter is approximately $6\times10^9$ cm$^{-2}$, and the

summer minimum is approximately $1\times10^9$ cm$^{-2}$. At low latitudes between 0 and 30°, a small semi-annual variation in the column abundance in seen with maxima of $5\times10^9$ cm$^{-2}$ in May and October. The stellar occultation measurements made by the GOMOS instrument onboard the Envisat satellite also showed a clear semi-annual oscillation in Na vertical column density at low latitudes that merges into an annual variation above 30° latitude (Fussen et al., 2010; Langowski et al., 2017). In the present study, the latitude dependence of the seasonal variation in the Na layer from the meridional chain of the CMP Na lidars

is discussed in section 3.2.

    Figure 4 shows the geographical distribution of the annual mean Na column number density at latitudes below 40° in a 5°$\times$ 5° grid, based on the six-year OSIRIS measurements from 2004 to 2009. As in Figure 3, grey shading indicates the region where there are no measurements or not enough measurements at mid- to high-latitudes in winter to calculate the annual average. The annual average Na column at low-latitudes peaks in two broad bands between 10°–40°N and S, with

pronounced areas of high concentration over eastern Asia and the north Pacific, the north Atlantic, and the south Pacific and





south Atlantic. This geographical distribution is fairly consistent with the distribution of Es layers reported by Yu et al. (2019b). The observations of the Na layers have been made in most of these regions using ground-based lidars at Arecibo, Puerto Rico (18.3°N, 67.0°W) (Cai et al., 2019b); Cerro Pachón, Chile (30.3°S, 70.7°W) (Liu et al., 2016); Uji, Japan (34.9°N, 135.8°E) (Suzuki et al., 2010); Beijing, China (40.2°N, 116.2°E) (Gong et al., 2013); Hefei, China (31.8°N, 117.3°E) (Dou et al., 2009);

Wuhan, China (30.5°N, 114.4°E) (Yi et al., 2013); and Haikou, China (19.5°N, 109.1°E) (Jiao et al., 2017). The limb-scanning radiance data from OSIRIS provide a near-global view of Na layers. However, because the peak Na density was not observed during the dark winter months, the annual mean Na column abundances at high latitudes from OSIRIS cannot be made.

The maps in Figure 5 show the global distributions of the Na number column density from OSIRIS for four different seasons in a $5° \times 5°$ grid. The Na layer clearly shows a global seasonal dependence with a pronounced minimum in the

summer hemisphere. The Na column density at high latitudes reaches over $5 \times 10^9$ cm$^{-2}$ in autumn and winter, and decreases to $1 \times 10^9$ cm$^{-2}$ in summer. High concentration above $4.5 \times 10^9$ cm$^{-2}$ in all four seasons can be seen over eastern Asia and the north Pacific, the north Atlantic, and the south Pacific and south Atlantic. Note that grey-shaded areas represent areas where OSIRIS limb-scanning radiance measurements were not made during the winter at high latitudes. Whereas, the ground-based lidar measurements are a supplement to satellite data sets when the high-latitude Na layers are not solar-illuminated in winter.

The Na column density has a maximum in winter from worldwide lidar observations (Plane et al., 1999; Yuan et al., 2012; Li et al., 2018) and global modeling studies of meteoric Na in the MLT (Marsh et al., 2013; Wu et al., 2021). The lower panels during each season depict the northern and southern polar views of the Na column density, and these views make the summer minimum in the high-latitude Na layers clearer. The significant summer-time depletion of high-latitude Na layers is primarily attributed to the very low temperature at the summer mesopause because of dramatic adiabatic cooling of upwelling air (Plane,

2003) and secondarily attributed to efficient removal of metallic species on noctilucent cloud particles (Plane et al., 2004; Raizada et al., 2007; Plane, 2012).

To compare with the OSIRIS results, the global map of the annual mean Na column number density simulated by WACCM-Na from 2004 to 2009 is shown in Figure 6. The WACCM simulated Na column density ranges from $3.0 \times 10^9$ cm$^{-2}$ to $4.6 \times 10^9$ cm$^{-2}$. The largest Na column densities are seen at 60°S high-latitude, where the column density is $4.7 \times 10^9$ cm$^{-2}$.

In the tropics, the Na column is relatively small around $3.0 \times 10^9$ cm$^{-2}$. The WACCM-Na model prediction has been shown in good agreement with the Na column density measured by the mid-latitude lidar at Fort Collins (41°N, 104°W) and the lidar at the South Pole (Marsh et al., 2013).

The maps in Figure 7 show the global distributions of the WACCM simulated Na column density for four different seasons during the period 2004–2009. The Na column density in WACCM-Na shows a similar seasonal variation to the OSIRIS ob-

servations in Figure 5. At high latitudes, the summer minimum is around $2.0 \times 10^9$ cm$^{-2}$ in the WACCM-Na model, which is consistent with the measurements from OSIRIS in summer. The winter maximum in WACCM-Na is $5.0 \times 10^9$ cm$^{-2}$ at high latitudes. However, the ares of pronounced Na column density observed from OSIRIS shown in Figures 4–5, e.g. in eastern Asia and the north Atlantic, are not reproduced in the WACCM-Na simulations. In the following section, the ground-based measurements of Na layers from four lidars in the CMP are compared to satellite observations and WACCM simulations.





### 3.2 Mid-latitude and low-latitude Na layer from the Odin satellite, ground-based lidars, and the WACCM-Na model

Figure 8 shows the time series of mean Na number density profiles in a 5 day× 2 km grid from the Na lidars at Beijing (40.2°N, 116.2°E), Hefei (31.8°N, 117.3°E), Wuhan (30.5°N, 114.4°E), and Haikou (19.5°N, 109.1°E). The lidar measurements at the four stations, particularly at the Beijing and Hefei mid-latitude stations, show an annual variation in Na layers with a winter maximum (Figure 8a–b). There are over 9,000 hours and 5,000 hours observations of the Na layer at Beijing and Hefei, respectively, and more than 1,800 hours at the other two stations.

Figure 9a shows the comparison of annual mean Na column density between OSIRIS measurements and Na reference by Plane (2010). Good agreement is found between 20°S and 40°N latitudes. Whereas, the annual mean OSIRIS at higher latitudes is largely underestimated due to less measurements in winter. Therefore, the height-latitude distributions of the neutral Na layer from OSIRIS between 20°S and 40°N latitudes and Na lidars are shown in Figures 9b and c. In Figure 9b, the annual mean Na number density from OSIRIS shows a northern number density over 3000 cm$^{-3}$ at 10°N–40°N latitudes and a southern density over 3000 cm$^{-3}$ at 15°S latitude. Figure 9c shows the Na number density from the lidars at Beijing (40.2°N), Hefei (31.8°N), Wuhan (30.5°N), and Haikou (19.5°N), along with superposed contour lines of Na number density from OSIRIS. The peak number density of Na layers at the four stations is between 2500 cm$^{-3}$ and 3000 cm$^{-3}$, in accord with the observations from OSIRIS. The low-latitude Na layer at Haikou has a relatively higher peak density of nearly 2900 cm$^{-3}$ than the mid-latitude Na layers at Hefei and Beijing with the peak density of 2500–2550 cm$^{-3}$. The Na layer at Wuhan has a moderate peak density of approximately 2750 cm$^{-3}$. The Na number densities at the four lidar stations agree well with the latitudinal variation in the Na number density observed from OSIRIS. A large Na column density in eastern Asia at 10°N–40°N latitude can also be found in the global map of the Na layer from OSIRIS over four seasons (Figures 4 and 5), which is not reproduced by WACCM-Na (Figures 6 and 7).

More recently, the distribution of Mg$^+$ column density simulated by WACCM-X (Wu et al., 2021) shows a stronger convergence along the magnetic equator. The equatorial metallic ions are uplifted by V×B forcing and then drift down along the magnetic field lines to the subtropical region. The transport of metallic ions is generally consistent with the fountain effect and the fountain effect is stronger over east Asia (Wu et al., 2021). The metal layer can be influenced by the metallic ions as the major reservoir through the recombination between ions and electrons, since the neutral atoms are not directly transported by the electromagnetic field (Cai et al., 2019a). It could explain why WACCM-Na does not capture the large Na observed by the CMP lidars at 10°N–40°N.

Figure 9d shows the height-latitude distribution of Es layers represented by the S4max from COSMIC satellites. The latitude distribution of Es layers show a north-south hemisphere asymmetry. A broad maximum between 10°N and 40°N in the Northern Hemisphere, and a smaller peak distribution around 15°S latitude in the Southern Hemisphere. A small enhancement of the Es layer intensity can be found at 75°N–90°N northern high latitudes.

It is an interesting feature that both the neutral Na layer and Es layer intensity have a similar latitude distribution (at least below 40°N latitude). Many previous studies have shown a strong correlation between local SSLs and Es layers (Dou et al., 2010; Kane et al., 2001; Sarkhel et al., 2012; Dou et al., 2013). The neutralization of metallic ions is the most commonly





accepted mechanisms for SSL formation (Cox and Plane, 1998). Na layer and Es layers both have a prominent seasonal

variation. Na layer has a maximum density in winter and the Es layers have a maximum density in summer. The geographical

distributions of Na atoms and metallic ions with Es layers, particularly the similar pronounced areas of high concentration

should be related. Therefore, it indicates that it is necessary to incorporate the electrodynamical transport of metallic ions

within Es layers into a global model of Na.

Previous ground-based measurements at 51°N (Gibson and Sandford, 1971), 41°N (She et al., 2000; Yuan et al., 2012),

23°S (Andrioli et al., 2020), and 90°S (Gardner et al., 2005) have reported that the Na layers display a seasonal variation

at altitudes greater than 20°. A semi-annual oscillation in the Na layer was found from the recent lidar observations in the

Southern Hemisphere from a dual Na/K Na lidar at São José dos Campos (23.1°S, 45.9°W), Brazil (Andrioli et al., 2020), in

which the seasonal variation in meteor ablations can not account for the semi-annual behavior of Na layers.

For further comparison with the CMP Na lidar observations, the measurements of Na layers from OSIRIS made within

a region of $\pm$ 5° latitude and longitude square centred on each ground-based lidar station were used. The measurements of

Na number density in the morning from OSIRIS at about 6:00 LT (descending node) are compared with the ground-based

observations from Na lidars at 4:00–6:00 LT. The measurements in the evening from OSIRIS at about 18:00 LT (ascending

node) are compared with the observations from Na lidars at 18:00–20:00 LT.

Figure 10 shows the monthly variations in Na number density in the morning from OSIRIS and the Na lidars at Beijing,

Hefei, Wuhan, and Haikou. The Na number densities from OSIRIS and Na lidar both show a significant seasonal variation at

Beijing in Figures 10a and b. The Na number density from OSIRIS shows a larger summer minimum of 2200 cm$^{-3}$ with the

standard error of 200 cm$^{-3}$ in June and a winter maximum of 5200 cm$^{-3}$ with the standard error of 400 cm$^{-3}$ in October

than that from the Na lidar with a summer minimum of 1800$\pm$200 cm$^{-3}$ in July and a winter maximum of 3300$\pm$100 cm$^{-3}$

in December. The standard error is calculated from the standard deviation of monthly mean divided by the square root of the

number of points in each month. Figures 10 c and d show Na number density from OSIRIS and Na lidar at Hefei. The summer

depletion of the Na layer is observed at Hefei from lidar, with a minimum of 1700$\pm$100 cm$^{-3}$ in July. The maximum Na

density in winter at Hefei is 3100$\pm$100 cm$^{-3}$ in November from Na lidar and 5000$\pm$400 cm$^{-3}$ in October from OSIRIS. The

depletion of the relatively low-latitude Na density in summer is not significant at Wuhan and Haikou, and the Na layers show

a semiannual variation with peaks at equinox from the Na lidars. In Figure 10f, the winter maximum from Na lidar at Wuhan

is 4100$\pm$300 cm$^{-3}$ in January and the secondary peaks are 3000$\pm$300 cm$^{-3}$ in April and 3700$\pm$200 cm$^{-3}$ in November. In

Figure 10h, the peaks are 4300$\pm$200 cm$^{-3}$ in October and 3400$\pm$300 cm$^{-3}$ in March. In Figures 10e and g, the Na number

densities from OSIRIS show an annual variation with maxima of 5400$\pm$400 cm$^{-3}$ and 5600$\pm$700 cm$^{-3}$ at Wuhan and Haikou

in October. The Na number density from OSIRIS is generally larger than that from Na lidars at the four stations.

On the right panels of Figure 10, the peak densities of the Es layers in summer are superimposed on the Na density contour

from lidars, represented by the S4max at levels of 0.6, 0.8, 1.0 as the light to dark blue contour lines. The response of the Na

layer to the strong Es layers in summer is not very significant. Even though the density of the Na layer is lowest in summer

because of the very low temperature at the mesopause, the Na layer can be influenced by the neutralisation of metallic ions

with a summer maximum of Es layers. It has been found that the sporadic enhanced Na layers in the upper mesosphere and



thermosphere occur more frequently in summer than around the equinoxes, which are generally associated with the presence

of the Es layer (Dou et al., 2013). The sudden enhancement of neutral metal density, which appears abruptly in a thin layer (full width at half maximum ∼1 km) on the topside of the normal metal layer (Clemesha, 1995), has a strong seasonal dependence with the highest occurrence rate in summer but it was rarely observed in other seasons (Qiu et al., 2016). Based on the observations from lidars and satellites, the climatology and seasonal variations of the Es layers, the sporadic metal layers (SSLs for Na), and the Na layers are analysed later.

Figure 11 shows the monthly variations in Na number density in the evening from OSIRIS and Na lidars. Because the time of the ascending node shifted from 18:00 LT to 18:50 LT over the years, the number of Na density data in the evening, particularly in the equatorial region, is much less than that in the morning from the OSIRIS limb-scanning radiance measurements (Hedin and Gumbel, 2011). The Na number density in the evening is smaller than that in the morning. The Na number density from OSIRIS is generally larger than that from the Na lidars. In the right-hand panels of Figure 11, the observations from the four

lidars show a significant summer depletion of the Na layer. The observations from Na lidars show a shift from an annual variation in Na density at Beijing and Hefei to a semiannual variation at Wuhan and Haikou, which is consistent with the seasonal variation with latitude in Figure 10. The Na number density minimum is around 1500 cm$^{-3}$ in summer, and the maximum density is around 2500 cm$^{-3}$.

Figure 12 shows the monthly variations in the WACCM simulated Na number density at 6:00 LT in the morning and 18:00 LT

in the evening. In general, WACCM-Na is capable of reproducing the seasonal variation in the Na layer. However, WACCM underestimates the seasonal variability of the Na layer as observed by lidars at Wuhan and Haikou. It may arise from the fact that the seasonal variablity of vertical transport of atmospheric constituents modeled in WACCM is underestimated since gravity waves and turbulence fluctuations are not included or the equatorial plasma fountain effect is not included in model. The vertical flux of Na atoms in the mesopause region employed in WACCM-Na is much smaller than that from lidar measurements

(Marsh et al., 2013). The WACCM simulations at 6:00 LT show a peak Na number density of approximately 3000 cm$^{-3}$ in October and a secondary peak of approximately 2500 cm$^{-3}$ in March. The WACCM simulated Na density at 18:00 LT shows a similar variation. The peak density is approximately 3000 cm$^{-3}$ in October, whereas the secondary peak in March is not significant.

Figure 13 shows the full diurnal variation of the WACCM simulated Na number density and the lidar-observed night-time

Na number density. The Na layer is enhanced after midnight in the WACCM simulations. It is consistent with the observations from Na lidars that the morning Na density around 06:00 LT is 10 %–30 % larger than the evening Na density around 18:00 LT. After sunrise, an increase of Na density on the bottom side of the layer and removal of Na density on the topside of the layer can be found in WACCM simulations. Ion-molecule chemistry plays an important role in the Na layer above 96 km altitude (Cox and Plane, 1998). The influence of the odd oxygen (O and O$_3$) /hydrogen (OH, HO$_2$) chemistry and photochemistry of

the major reservoir species (NaHCO$_3$) dominates below 96 km altitude (Plane et al., 1999, 2015; Yuan et al., 2019; Xia et al., 2020). Due to the increased solar radiation and the increased NO$^+$ and O$_2^+$ ions, atomic Na is removed off the topside of the Na layer after sunrise and transformed to Na$^+$. The photolysis of the main reservoir, NaHCO$_3$, causes an increase in Na atoms on the bottom side of the layer after sunrise. WACCM also reproduces the diurnal variation in the Na layer driven by the diurnal





tidal modulations (Liu et al., 2013; Yuan et al., 2014a). The right-hand panels of Figure 13 show that lidar-measured Na density

after midnight is larger than in the evening, in agreement with WACCM.

Figure 14 compares the seasonal profiles of Na number density from OSIRIS, Na lidars, and WACCM-Na at Beijing (a–c), Hefei (d–f), Wuhan (g–i), and Haikou (j–l). The left panels show the vertical profiles of Na number density from OSIRIS with a 2 km altitude resolution between 76 and 106 km, within a region of $\pm\,5°$ latitude and longitude square centred on each lidar station. Due to the absence of daylight when the solar zenith angle is larger than $92°$, there are fewer observations in winter

than in the other three seasons. The observations from OSIRIS show similar seasonal mean vertical profiles of Na density as above the four lidar stations. The maximum Na density is 4500–4900 $cm^{-3}$ in autumn, with a peak height around 91 km. The peak density is 2700–3500 $cm^{-3}$ during other seasons.

The seasonal profiles of night-time Na number density from the Na lidars are shown in the middle panels. In Figures 14b and h, the Na layer has a significant seasonal dependence at Beijing and Wuhan, with the maximum Na density of approximately

3000 $cm^{-3}$ in winter and a peak height of 91 km and 93 km, respectively. In Figures 14e and k, the maximum Na density at Hefei and Haikou is 2800 $cm^{-3}$ and 3300 $cm^{-3}$ in autumn, with a peak height of 91 km. Although the vertical resolution of the profiles of Na number density from OSIRIS limb-scanning radiance measurements is lower than that of ground-based Na lidars, the OSIRIS data are nevertheless in good agreement with the measurements from the Na lidars. The observations from the OSIRIS spectrometer on the Odin satellite therefore provide a reliable measurement of the Na layer. The right-hand

panels show the seasonal profiles of Na number density simulated by WACCM-Na, which exhibit similar seasonal profiles. The maximum Na density is 2800-3000 $cm^{-3}$ in autumn, with a peak height of 88 km. The peak density is 2300–2500 $cm^{-3}$ during the other seasons.

In Figure 15, the monthly variations in the climatology of Na column number density from OSIRIS and Na lidars, and the Es layer intensity represented by the S4max from COSMIC are compared to the monthly variations in the probability of SSL

events observed by the four Na lidars in 2011/2012 reported in Dou et al. (2013). The dark and light green lines represent the monthly mean Na column number density from Na lidars and OSIRIS, respectively, with the standard errors of the mean represented by the width of the dark green bars and the light green shaded areas. The standard error was calculated from the standard deviations of the monthly mean divided by the square root of the number of points in each month. The orange and yellow bars in the histogram represent the monthly variations in the number of SSL nights and total nights measured by lidars.

The orange line represents the probability of SSLs per day. The intensity of the Es layer at the height of peak Na density around 92 km is represented by the blue lines, with the standard errors represented by the width of the blue shaded areas.

Figure 15a shows a strong correlation between Es layers (blue line) and SSLs (orange line) at Beijing, with a summer maximum and a winter minimum. The probability of SSLs can reach up to $100\,\%$ per day (ten SSL events out of ten nights) in June, compared to approximately $6\,\%$ per day in December and January. The intensity of the Es layer is represented by the

S4max index (Yu et al., 2019b, 2021b; Qiu et al., 2021a). S4max is 0.96 in June, more than three times the value of 0.27–0.28 in December and January. Both the observations from OSIRIS and the Na lidar show an annual change in the Na column number density, with a summer minimum and a winter maximum. The SSLs are more correlated to Es layers at Beijing at a higher latitude compared the other three stations. In Figures 15b–d, the probability of SSLs shows a semi-annual variation at Hefei,



Wuhan, and Hefei. The probability of SSLs reaches a peak in summer and reaches a secondary peak in February and March.

The summer maximum occurrence of SSLs is consistent with the summer maximum intensity of the Es layer. The secondary peak in occurrence of SSLs in February and March is consistent with the climatology of Na layers from lidaras. Yu et al. (2021c) found a winter-to-summer dynamical process of the Es layer induced by the lower thermospheric meridional circulation. The dynamical process such as the meridional transport of metallic ions with the influence of atmospheric meridional circulations is likely related to the monthly variations in the occurrence of SSLs. The formation of low-latitude Es layers is attributed to

echoes from the equatorial electrojet irregularities (Whitehead, 1989). The Es layers at low latitudes are more affected by the recurrent geomagnetic activity rather than the dynamical process caused by the background wind field (Whitehead, 1970; Yu et al., 2021a). Therefore, the correlation between the probability of SSLs and the density of Es layers at low-latitude Haikou station in Figure 15d is not as significant as those at mid-latitudes in Figures 15a–c.

## 4  Summary

We present a study on the climatology of Na layers at mid- and low-latitudes, from the ground-based measurements obtained from a lidar network of four Na fluorescence lidars along 120°E longitude in the CMP, the OSIRIS limb-scanning radiance measurements detected by the Odin satellite, and WACCM-Na simulations.

The geographical distribution of the Na layer shows a global seasonal dependence from OSIRIS measurements and WACCM model. The pronounced areas of high Na concentration above $4.5 \times 10^9$ cm$^{-2}$ can be seen over eastern Asia and the north Pa-

cific, the north Atlantic, and the south Pacific and south Atlantic from OSIRIS in all four seasons. This geographical distribution is fairly consistent with the distribution of Es layers reported by Yu et al. (2019b).

Up to January 2021, the Na lidars at Beijing (40.2°N, 116.2°E), Hefei (31.8°N, 117.3°E), Wuhan (30.5°N, 114.4°E), and Haikou (19.5°N, 109.1°E) collected vertical profiles of Na density for a total of 2,136 nights (19,587 h). Ground-based Na lidars can provide extremely high temporal and altitude resolution local observations of Na layers. However, the lidar can

only provide point measurements. The space-based observations add to the restricted coverage of ground-based equipment by providing a global climatology of Na layers. The OSIRIS limb-scanning radiance measurements were not taken during the dark winter months owing to a lack of sunshine when the solar zenith angle is larger than 92°. The CMP Na lidars provide high temporal and vertical resolution measurements of local Na layers at mid- and low-latitudes along 120°E longitude as a supplement to space-based observations, which can present a test of OSIRIS and WACCM.

Good agreement is found between lidar observations, satellite measurements, and WACCM simulations. At mid-latitudes and high latitudes between 30° and 90°, the Na layers from OSIRIS show a significant annual variation, with a winter maximum and a summer minimum. At low latitudes between 0 and 30°, a semi-annual variation is found. In accord with the measurements from OSIRIS, the observations of Na number density from the four lidars show a significant annual variation at Beijing and Hefei and a semi-annual variation at Wuhan and Haikou. The Na number density from OSIRIS is slightly larger

than that from Na lidars at the four stations, particularly in autumn and early winter as a result of significant uncertainties in Na density retrieved from much less satellite radiance measurements. WACCM underestimates the seasonal variability of Na



layers observed at Wuhan and Haikou lidar stations. This discrepancy suggests the seasonal variability of vertical transport of constituents is underestimated in WACCM because much of gravity waves is not resolved in model. The close relationship between the ionospheric Es layer, the Na layer, and SSLs indicates that it is important to consider the dynamical and electrody-
namical processes of metallic ions in the lower E region of the ionosphere coupling with the Na layer into a global atmospheric model of metals.

*Data availability.* The Na lidar data are available from the Data Centre for Meridian Space Weather Monitoring Project (https://data. meridianproject.ac.cn/data-directory/) and the National Space Science Data Center, National Science & Technology Infrastructure of China (http://www.nssdc.ac.cn). OSIRIS data are available at https://research-groups.usask.ca/osiris/data-products.php. The COSMIC satellite ra-
dio occultation data are available from the CDAAC website (https://data.cosmic.ucar.edu/gnss-ro/).

*Author contributions.* BY designed the study and wrote the manuscript. XX and CJS supervised and provided support and suggestions on an early version of the manuscript. MJ processed Na layer data from the raw photon count profiles observed by Na fluorescence lidars. WF, JMCP, and DRM contributed to the WACCM-Na simulations. JH and JG provided the global Na density retrieved from Odin/OSIRIS. XD contributed to the discussion of the results and the preparation of the manuscript. All authors discussed the results and commented on the
manuscript at all stages.

*Competing interests.* The authors declare that they have no conflict of interest.

*Acknowledgements.* We acknowledge the Chinese Meridian Project, the Solar-Terrestrial Environment Research Network (STERN), and the National Space Science Data Center, National Science & Technology Infrastructure of China for providing the Na lidar data. The authors acknowledge the Odin satellite mission for providing the global measurements of Na layers and the Constellation Observing System for
Meteorology, Ionosphere, and Climate (COSMIC) Data Analysis and Archive Center (CDAAC) for providing the COSMIC radio occultation data. The authors would like to thank the National Science & Technology Infrastructure of China. This work has been supported by the B-type Strategic Priority Program of CAS (grant no. XDB41000000), the National Natural Science Foundation of China (grant Nos. 42125402,42188101, 41974174, 41831071), the Project of Stable Support for Youth Team in Basic Research Field, CAS (grant No. YSBR-018), Anhui Provincial Natural Science Foundation (grant no. 1908085QD155), and the Fundamental Research Fund for the Central
Universities. Wuhu Feng and John M. C. Plane were supported by the Natural Environment Research Council (grant no. NE/P001815/1). Bingkun Yu was supported by the Royal Society for the Newton International Fellowship (grant no. NIF\R1\180815).





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





## Figures & Tables

**Table 1.** Na lidar measurements and main parameters of the Na lidars

|  | Beijing | Hefei | Wuhan | Haikou |
|---|---|---|---|---|
| Location | (40.2°N, 116.2°E) | (31.8°N, 117.3°E) | (30.5°N, 114.4°E) | (19.5°N, 109.1°E) |
| Data coverage | 2010–2018 | 2005–2021 | 2011–2018 | 2010–2018 |
| Number of nights | 906 | 682 | 186 | 362 |
| Number of hours | 9583 | 5052 | 1865 | 3087 |
| Spatial resolution | 96 m | 150 m | 96 m | 96 m |
| Time resolution | 3 min | 4 min | 5 min | 3 min |
| Transmitter |  |  |  |  |
| Pulse energy | 40 mJ | 60 mJ | 60 mJ | 42 mJ |
| Pulse width | 10 ns | 6 ns | 6 ns | 10 ns |
| Line width | 1.5 GHz | 1.5 GHz | 1.5 GHz | 1.5 GHz |
| Receiver telescope |  |  |  |  |
| Diameter | 1 m | 1 m | 0.5 m | 1 m |
| Receiver filter | 589 nm | 589 nm | 589 nm | 589 nm |
| Bandwidth | 1 nm | 1 nm | 1 nm | 1 nm |



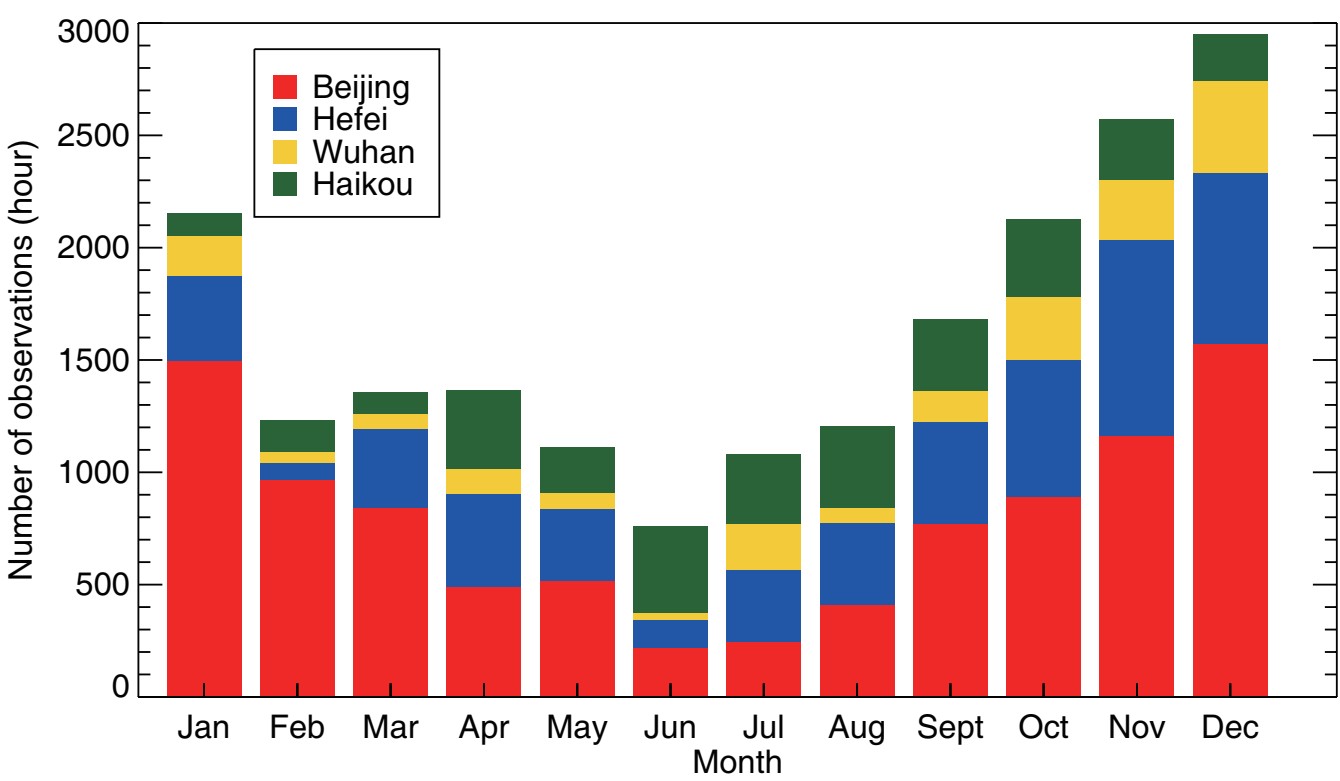

**Figure 1.** Monthly variations in the number of Na observations measured by the four Na lidars during the period 2005 to 2021.





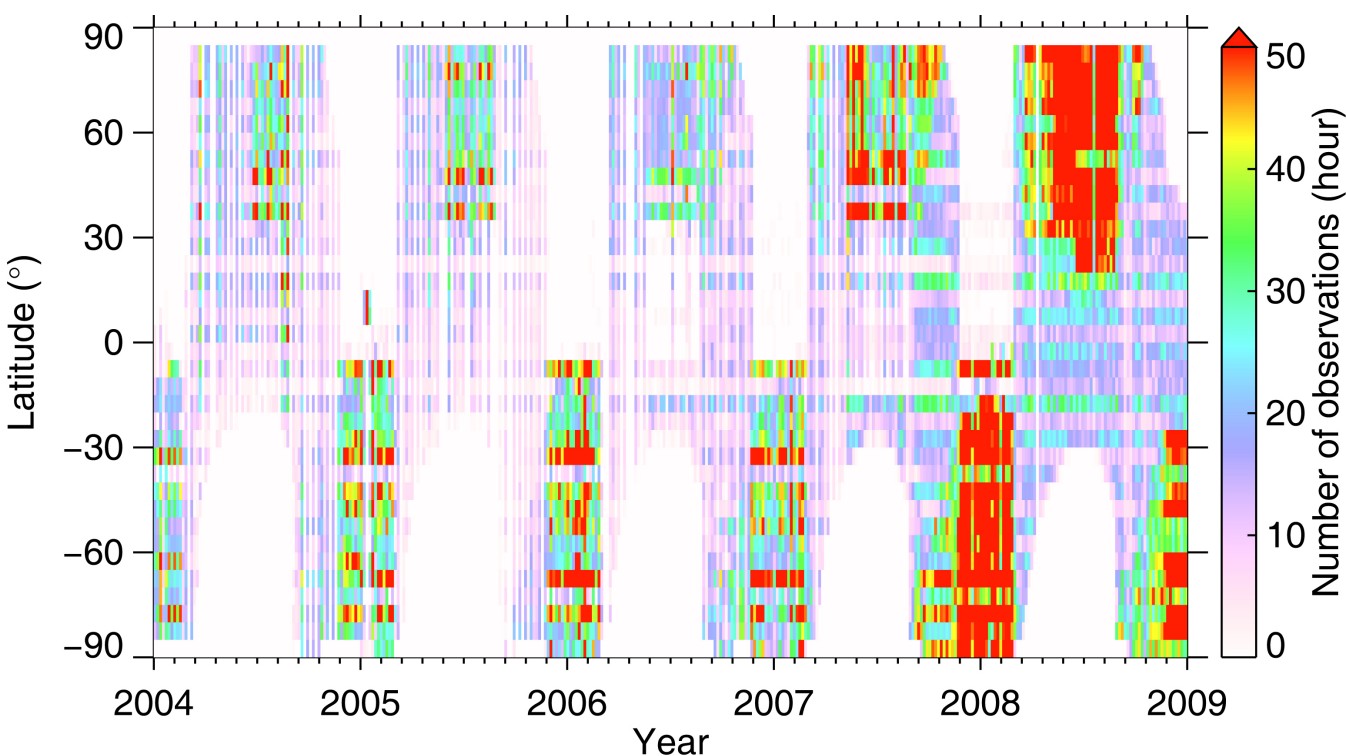

**Figure 2.** Time-latitude distribution of the number of the OSIRIS limb-scanning radiance measurements made by the Odin satellite during the period 2004 to 2009, with a resolution of a 5 day × 5° latitude grid.




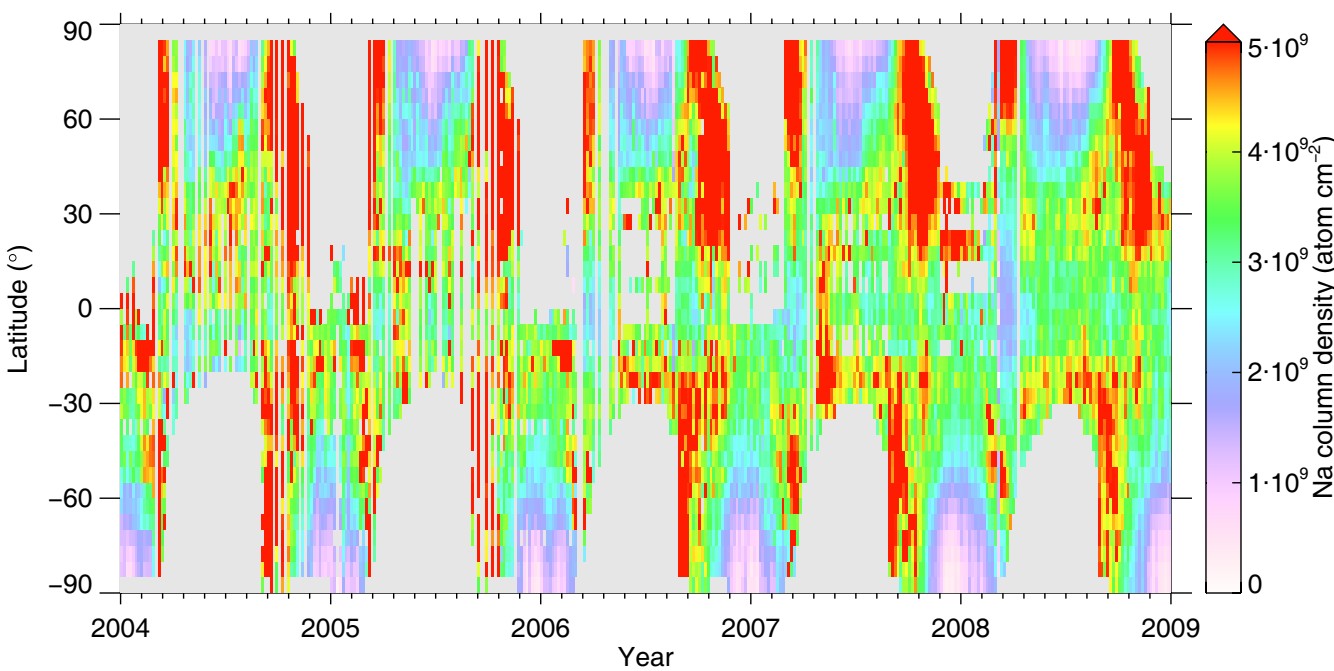

**Figure 3.** Time-latitude distribution of the Na column number density from the OSIRIS limb-scanning radiance measurements made by the Odin satellite, with a resolution of a 5 day × 5° latitude grid. Na column density is integrated from 70 to 120 km altitude. Grey-shaded areas indicate the regions where OSIRIS limb-scanning radiance measurements were not made in winter.





**Figure 4.** Geographical distribution of the annual mean Na column number density below 40° latitude from the OSIRIS limb-scanning radiance measurements made by the Odin satellite during the period 2004 to 2009, with a spatial resolution of a 5° × 5° grid. Grey shading indicates the regions where there are no measurements or not enough measurements at mid- and high-latitudes in winter to calculate the annual average.







**Figure 5.** Seasonal variations in the Na column number density from the OSIRIS limb-scanning radiance measurements made by the Odin satellite during the period 2004 to 2009, with a spatial resolution of a $5° \times 5°$ grid. Maps for (a) autumn (September, October, November), (b) winter (December, January, February), (c) spring (March, April, May), and (d) summer (June, July, August). It is worth emphasising that the grey-shaded areas represent the areas where the OSIRIS limb-scanning radiance measurements were not made during the dark winter months at high latitudes, but these areas should have a winter maximum of Na column number density as indicated by the seasonal variations in Na layers from the ground-based lidar observations.



**Figure 6.** Global geographical distribution of the annual mean WACCM simulated Na column number density during the period 2004 to 2009.



**Figure 7.** Seasonal variations in the WACCM simulated Na column number density during the period 2004 to 2009. Maps for (a) autumn (September, October, November), (b) winter (December, January, February), (c) spring (March, April, May), and (d) summer (June, July, August).





**Figure 8.** Time series of mean Na number density in a 5 day × 2 km grid from Na lidars at (a) Beijing (40.2°N, 116.2°E), (b) Hefei (31.8°N, 117.3°E), (c) Wuhan (30.5°N, 114.4°E), and (d) Haikou (19.5°N, 109.1°E).





**Figure 9.** (a) Annual mean Na column density from OSIRIS and Na reference by Plane (2010). Height-latitude distribution of the neutral Na layers and metallic ions: (b) Na number density from OSIRIS between 20°S and 40°N latitudes; (c) Na number density from the lidars at Beijing (40.2°N), Hefei (31.8°N), Wuhan (30.5°N), and Haikou 28 9.5°N), with superposed contour lines of Na number density from OSIRIS; (d) Es layers represented by the S4max from COSMIC satellites.





**Figure 10.** Comparisons of monthly mean of Na number density in the morning, from OSIRIS at about 6:00 LT (descending node) and Na lidars at 4:00–6:00 LT: (a–b) at Beijing, (c–d) at Hefei, (e–f) at Wuhan, and (g–h) at Haikou. The summer maximum Es layers represented by S4max at levels of 0.6, 0.8, 1.0 are shown as light to dark blue contour lines, superposed on the Na density contour from lidars on the right pannels.







**Figure 11.** Comparisons of monthly mean of Na number density in the evening, from OSIRIS at about 18:00 LT (ascending node) and Na lidars at 18:00–20:00 LT: (a–b) at Beijing, (c–d) at Hefei, (e–f) at Wuhan, and (g–h) at Haikou.





**Figure 12.** Monthly mean of the WACCM simulated Na number density in the morning at 6:00 LT and in the evening at 18:00 LT: (a–b) at Beijing, (c–d) at Hefei, (e–f) at Wuhan, and (g–h) at Haikou.







**Figure 13.** Diurnal variations of the WACCM simulated Na number density and the lidar-observed nighttime Na number density: (a–b) at Beijing, (c–d) at Hefei, (e–f) at Wuhan, and (g–h) at Haikou.



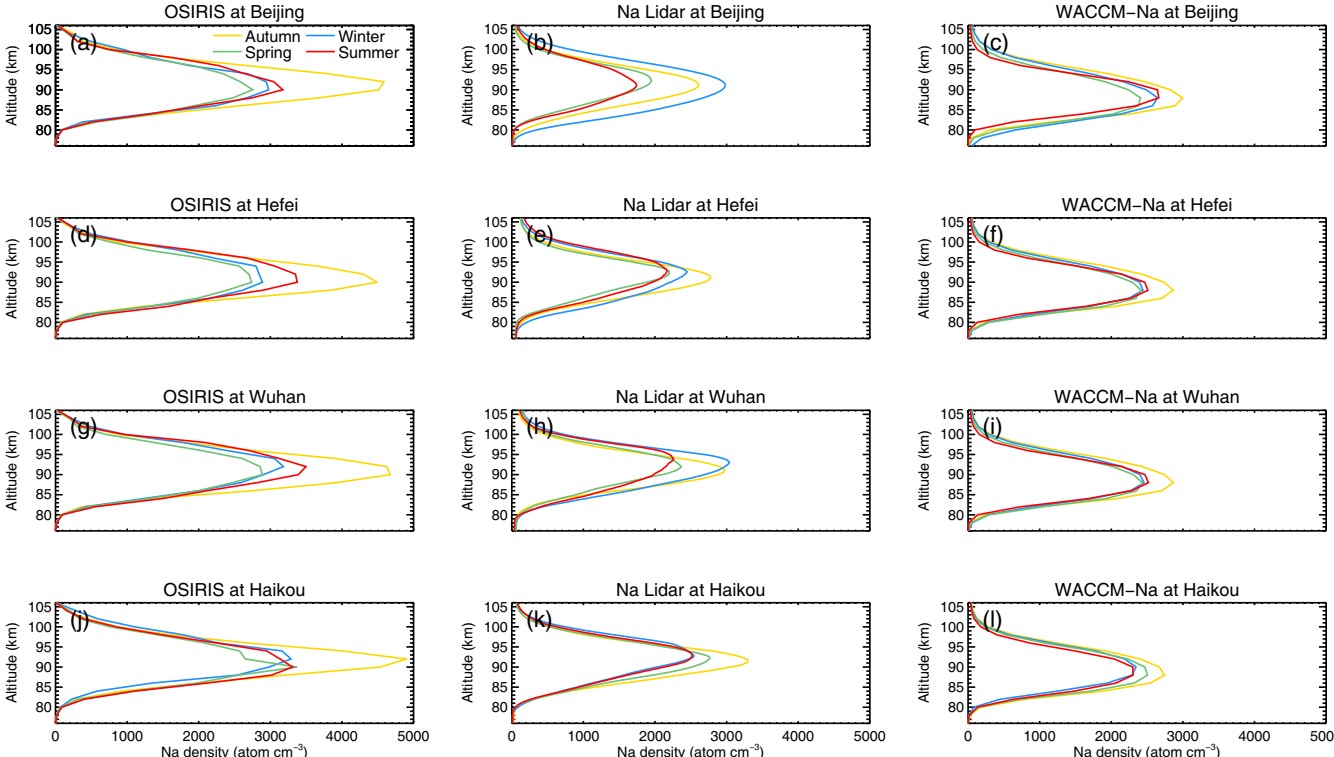

**Figure 14.** Comparisons of seasonal Na number density profiles from OSIRIS, Na lidars, and WACCM: (a–c) at Beijing, (d–f) at Hefei, (g–i) at Wuhan, and (j–l) at Haikou.





**Figure 15.** Monthly variations in probability of SSLs per day (orange lines), Es layer intensity represented by S4max at the height of peak Na density around 92 km (blue lines), and Na column number density between 76 and 106 km from OSIRIS (light green lines) and Na lidars (dark green lines) in 2011/2012, at (a) Beijing, (b) Hefei, (c) Wuhan, and (d) Haikou. The orange and yellow bars in the histogram exhibit the number of SSL nights and total nights measured by lidars. The standard errors of the metallic ion density are represented by the width of the blue shaded area. The standard errors of the monthly Na column number density from OSIRIS and Na lidars are represented by the width of the light green shaded area and the dark green bars. The standard error was calculated from the standard deviations of monthly mean divided by the square root of the number of points in each month.