# Peer review of "Comparison of middle- and low-latitude sodium layer from a ground-based lidar network, the Odin satellite, and WACCM-Na model"

_EGUsphere, 2022_

## Author Response (AR1)

**Reviewer #1 comments (RC1):**

This manuscript discusses Na-layer lidar observations taken at four stations in China: Beijing, Hefei, Wuhan, and Haikou. All lidar stations are around 110E and span the latitudes from about 19N to 40N. These dataset offers a unique opportunity to study Na-layer and its relation to dynamics, chemistry and electrodynamics. The lidar measurements are then compared to satellite observations and model simulations with WACCM-Na, a version of a climate model that includes. The main conclusions of the study are a general agreement between observations but with significant discrepancies at times attributed to the lack of space-borne observations during norther winter in the darkness; and, a general disagreement between the model on the observations in terms of variability. In particular, the authors suggest that model variability can be improved with a better representation of the spectrum of subgrid gravity waves.

Authors' response: We would like to thank the reviewer for the valuable comments and suggestions on our manuscript that help us improve our work. With respect to the comments of the reviewer:

A few comments/queries below:
Line 122: Please define what S4max (or S4 index) is for the non-experts.
Authors' response: Thank you for your comments. The definition of S4max has been included in the revision.
Changes: Line 129.

4 & 6. Please use the same contour min/max for ease of comparison.
Authors' response: revised.

Line 170. Could the horizontal resolution be the problem in resolving those fine structures/peaks?
Authors' response: It could be one reason, while the pronounced area of high concentration over eastern Asia and the north Pacific in Figure 4 (90°–180°E, 10°–40°N) is much larger than the horizontal resolution of WACCM. Another aspect is that the limited resolution can result in a portion of unresolved sub-grid waves.

Line 181: Is it sub-sampling by 5 days or is it a 5-day average?
Authors' response: a 5-day average.
Changes: Line 194.

Figure 8: I seem to detect a quasi-biennial oscillation. Any thoughts?
Authors' response: Thank you for your comments. It could be a result of the influence of lower atmospheric oscillation, while the annual variation is the most significant. As the lidar data accumulated, it will be a valuable study in the future to investigate the long-term periodic trends of Na layer from lidars in details.

Line 187: replace less with few
Authors' response: done.
Changes: Line 201.

Figure 9c: I almost missed the vertical bars: maybe add the station name for clarity?

Authors' response: Thanks for your suggestion. The station names have been included.

Changes: Figure 9c.

Line 203-206: I don't understand what is implied here. Is the suggestion that lack of field line transport is a potential issue for WACCM-Na? How's that so? I thought WACCM transports ions as part of the chemistry package.

Authors' response: The transport of the neutral and ionized metallic species in WACCM-Na is only driven by the eddy/molecular diffusion and winds in the same way as most active chemical species. The self-consistent electrodynamical transport of metallic ions is proven to have an important impact on the global distribution of metal ions and atoms simulated by WACCM-X (Wu et al., 2021), which could be associated with the unresolved structures of WACCM-Na at latitudes of 10°-40°N.

Line 268: The sentences need some clarifications. I think the authors want to say that the limited resolution of sub-grid processes could explain the lack of variability. I buy that. However, the sentence seems to indicate that WACCM is missing all gravity wave and turbulence. And I don't buy that.

Authors' response: Thanks for this suggestion. The sentence has been revised.

Changes: Line 284.

Figure 14: Isn't the seasonal variability also different between OSIRIS and CMP lidars?

Authors' response: In the revision, we compare the Na data under the similar condition in Figure 13 (~6 LT for OSIRIS, 4-6 LT for lidars, and 6 LT for WACCM). The seasonal variability between the three measurements is compared. The result at 18 LT is not shown as there are much fewer OSIRIS observations at 18 LT than at 6 LT.

Changes: Figure 13 in the revision.

Lines 349-351: The disagreement is more than slight. More importantly, how is it that fewer observations result in a high bias?

Authors' response: The description has been changed from 'lightly larger' to 'larger'.

The overestimated Na number density from OSIRIS relative to that from lidars and WACCM-Na is due to the much fewer observations in autumn and winter (less than 10 hour per 5 day*5° latitude grid indicated in the time-latitude distribution of the number of the OSIRIS measurements in Figure 2). The lack of space-borne observations during norther winter in the darkness limits the accuracy of seasonal variability of the OSIRIS Na profile retrievals.

Changes: Line 373.

**Reviewer #2 comments (RC2):**

This paper utilizes the ground-based observations of Na, together with low-earth-orbiting satellite, and a general circulation model to study Na behavior globally. They not only provide the seasonal and latitudinal variation of Na, but also carry out data-model comparison, and provide explanation on it. This type of work (using ground based observation, satellite observation and model together) has never been carried out before, and it shall be a potentially crucial contribution to the study of metal atoms in the MLT region. The paper shall be published after a minor revision is made.

Authors' response: We would like to thank the reviewer for the valuable comments and suggestions on our manuscript that help us improve our work. With respect to the comments of the reviewer:

Line 38-56, in this paragraph, the author introduces the previous studies of sporadic Na layer. However, the author shall acknowledge the work of Cai et al., 2019b, which propose crucial conclusion on the seasonal occurrence of sporadic Na that has not been realized by other previous papers. As argued by Cai et al., 2019b, the summer Nas occurrence is higher in mid and mid-low latitudes because: 1 Na main layer density is lower in summer, which makes the sporadic na layer easier to be observer in a weaker main layer background. While in the winter, main layer is denser, and thicker. Therefore, it is much difficult to observe sporadic Na layer. Another aspect is that there is almost no difference of the neutralization of Na+ to Na in winter and summer. And the wind shear with convergence can also be formed in both winter and summer But the wind shear convergence formed in winter is below 90 km with much stronger neutralization, which makes the Na+ transferred into Na and being overwhelmed in the main Na layer. These crucial conclusions shall at least be mentioned.

Authors' response: Thank you for your comment. The study has been included in the introduction. More discussion of the seasonal occurrence of sporadic Na mentioned in this piece of work has been included in Section 4 Discussion.
Changes: Lines 49 and 345-352.

In the introduction, the authors give a summary of the shortages of these previous studies. This summary is not very clear in the current introduction. After this, the author shall then introduce their advantages (such as the first-time (maybe this is not accurate) data-model comparison between Lidar and waccmx)
Authors' response: Thanks for your suggestion and it has been revised accordingly.
Changes: Lines 65-66.

When introducing the Na lidar, the author shall clarify whether the Na lidar has the Faraday filter that can allow diurnal measurements, or can only allow nocturnal measurements??
Authors' response: The CMP lidars provide night-time measurements since they do not operate in daytime.
Lines 89-90.

Line 88 monthly variation is not accurate, please correct into the monthly distribution
Authors' response: it has been corrected.

Changes: Lines 91.

Since the author introduce several observation method and models, I suggest the author use the subsection to introduce each of them. The whole data and method without subsection look a mess. For example, 2.1 Na Lidar in CMP. 2.2 Odin satellite 2.3 COSMIC-1 RO 2.4 WACCM-Na.

Authors' response: The subsection has been included in the revised manuscript.

Line 115-116 Qiu et al., 2018 is not about the ion-neutral coupling, it just point out the

Finally, the error and uncertainty of the observations shall be pointed out, not just mention the previous papers.

Authors' response: The reference is removed. The uncertainty of the observations has been included.

Line 128-129 is the result calculated by averaging the longitude mean? Please clarify

Authors' response: The Na column density is zonally averaged. We have clarified in the revision.

Changes: Line 140.

Line 132 add 'which is' before 'consistent with'

Authors' response: it has been corrected.

Changes: Line 143.

Lines 145 and 156-157 the author had better point out the longitude range, rather than just saying the certain geographic name. Since these name means large area.

Authors' response: The longitude range has been pointed out in the revision.

Changes: Lines 156-157.

Line 160 this is not true. There are only several local observations of Na. Therefore, the author shall correct it into 'several local ground observations'

Authors' response: corrected.

Changes: Line 172.

Line 163-164 the temperature is not directly related to Na. It is the neutral chemistry of Na, which depends on neutral temperature, that influence the final Na density. Please clarify here to avoid misunderstanding.

Authors' response: clarified.

Changes: Lines 176-177.

Line 201 note that the fountain effect and the drifting along the field line occur simultaneously (Balan et al., 2018). Please remove 'then'.

Authors' response: corrected.

Changes: Line 214.

Line 221 replace 'altitudes' with 'latitudes'. Also for these local observations, the author shall add these near line 160, which states the local observations of Na

Authors' response: corrected and added.
Changes: Line 235

In all, section 3.2 is too long, and makes reader overwhelmed with a lot of information. I strongly suggest the author divide the section 3.2 into several shorter sections. Such as section 3.3: diurnal variations 3.4 seasonal variations 3.5 correlation between Es and SSL.

Authors' response: Thanks for your helpful suggestion. We have revised accordingly.

The author has presented many results of Na from local observation, satellite observation and the WACCM-Na. However, what I have seen now is a pile of observation and comparison. The author shall add a discussion section to provide explanation on these observations. I think this is not difficult for the author to carry this out.

Authors' response: Thanks for your suggestion and the discussion section is included in 4 Discussion.

**Reviewer #4 comments (RC4):**

This paper reports the climatology of sodium (Na) layer with lidar observations from 4 stations (with data ranging from 362 nights to 906 nights) at the middle- and low-latitude along 120oE operated by the Chinese Meridian Project (CMP). The results were compared to those with Infra-Red Imager System (OSIRIS) spectrometer onboard the Odin satellite and a global model of meteoric Na in the atmosphere (WACCM-Na); they found general agreement with some explainable differences. They also present the observations of sporadic Na layer (SSL) and the difference in correlation to sporadic E layer (Es) between mid-latitude stations and low-latitude stations. I find the study scientifically meaningful, clearly written with good connections to the past research. The paper is suitable for ACP. As stated below, I have a couple of major and minor queries; they can be accounted for with minor revisions.

Authors' response: We would like to thank the reviewer for the valuable comments and suggestions on our manuscript that help us improve our work. With respect to the comments of the reviewer:

Major queries:

(1). Your brief description of 4 different methodologies used in Section 2 is appreciated. The aim is of course to help the readers to gain confidence on the reported describe physical quantities by the different methods used. Therefore, a clear connection between the "observed (calculated)" quantity and the quantity under study should be made in each case. Such descriptions would not be needed for a reader experienced in all 4 methods, not too many in our field. I think your descriptions on OSIRIS and WACCM-Na look good. In COSMIC, why is the S4max index a good

measure of Es strength? Please give the definition of the S4 index! In the case of Na lidar, you could say something like, "If laser pulses with a fixed line-shape function are tuned to a fixed frequency within the Na D2-resonant absorption line at 589.6 nm, the received induced fluorescence intensity is proportional to the emitting Na density". People usually report on the uncertainty of the lidar measured Na density by its associated photon noise. There are two more sources of uncertainties. First, most practitioners convert the received photon profile at MLT to Na density there by assuming the lidar signal at a lower altitude (at 30 km for example) is the result of Rayleigh scattering from a "standard" atmosphere. The ignorance of air density at the time of measurement gives rise to uncertainty in Na density, please see Fig. 8(c) of Reference 1. Another uncertainty comes from the fact that most "stabilized" transmitting lasers do not lock to an absolute frequency reference, the laser light with bandwidth of 1.5 GHz for example could jump around within the Na fluorescence spectrum (FWHM about 1.2 GHz at 200 K). Likely with any luck, these two uncertainties would be smaller than the photon noise uncertainty reported. However, unlike the photon noise uncertainty, these unknown uncertainties remain the same with more data. My guess is that for climatology study as in this paper, there is no real concern. Therefore you do not need to go into the details in this paper, but you should in my view note their existence. However, as the CMP lidar data accumulated, it will be a concern, should one look for small effects like the long-term trends. See p. 399 of Ref. 2, if interested.

Authors' response: Thank you for your comments.

In the revised manuscript, the definition of S4 and its correlation with ionosonde observations of Es layers are introduced. The description of Na lidar is also included in Section 2 according to the reviewer's suggestion: 'the received induced fluorescence intensity is proportional to the emitting Na density'.

We also appreciate the reviewer's comments on the existence of the Na density uncertainties. These related references have been included. I agree with the reviewer. The uncertainties (normalization to atmospheric density at a lower reference height & laser frequency drift) is not the real concern in our climatology study. However, it will be a concern in the long-term change studies of Na layer in the future as the CMP lidar data accumulated.

Changes: Lines 89 and 129.

(2). Please expand the last two paragraphs of discussion in Section 3.2 between lines 303 and 328 somewhat to make the discussion on seasonal variation of SSL derived from the 4 latitudes clearer. The SSL formation has been researched in the past three decades and different mechanisms have been proposed. With the publication of Yu et al. (2021a) and Yu et al. (2021c), the authors claim that they've finally understood the formation of SSL in midlatitudes as well as in the tropics. Unfortunately, unlike other sections, these two paragraphs are overly compressed and not as clear as they need to be. For example, why do you show the intensity of the Es layer at the Na layer peak around 92 km (blue line), while the crucial thermospheric meridional circulations appeared between 100 - 120 km? Also, are most of SSL events occurred at the Na layer peak?

Authors' response: Sorry for the unclear statement of Figure 15 in the last two paragraphs. In the revision, the two paragraphs are revised to state more clearly and put in Section 4 Discussion.

The main scope of the study is the global climatology of Na layer from the CMP lidars, the Odin/OSIRIS satellite, and the WACCM-Na. We found that the global distribution of Na from Odin/OSIRIS is fairly consistent with the distribution of Es layers. Therefore, we conducted further analysis of the SSL, Es, and Na climatology in Figure 15. It is preliminary result of the SSL not the conclusive evidence for the formation of SSL, based on RO observations of Es layer, lidar and Odin/OSIRIS observations of Na layer, and SSL events. The climatology of SSLs shows agreement with Es layers. We infer that the variations of Es layer, e.g., mid-latitude Es affected by the thermospheric meridional circulation (95-115 km) (Yu et al. 2021a) and low-latitude Es affected by the geomagnetic activity (Yu et al. 2021c) could be related to the SSL formation. However, more detailed studies are needed combined with the Na & Es observations and WACCM simulations. The Es layer intensity represented by S4max was at 96 km rather than 92 km by our mistake. It has been corrected in the revision. The S4max at the altitude around 96 km (95-97 km) is used to study the possible effect of metallic ions on SSL when the Es layer descends below 100 km under the influence of tides, since the peak altitude of SSLs is around 96 km (Fig. 5 of Dou et al., 2013). Furthermore, following another referee #2's comments, more discussion of the seasonal occurrence of SSL is also included in Discussion.

[Figure]

**Figure 5.** Histogram of the peak altitude of SSLs (black bars) and TeSLs (yellow bars) at (a) Beijing, (b) Hefei, (c) Wuhan, and (d) Haikou.

Changes: Lines 320-352.

Minor queries:

A. You mentioned "exceptionally high temporal and vertical resolution (usually 3 minutes and 100 m)" in line 54. Would you like to mention Reference 3, where the Na layer observed with "exceptionally" high-resolution (60 ms and 15 m) appears to be composed of many downward propagating SSLs. It would be fun to speculate the underline physics.

Authors' response: Yes, the high-resolution observations of the Na density real significant features of Na layers in greater detail. The study has been included in Introduction.
Changes: Lines 45-46.

B. Question about CMP lidar stations. You utilized data from 4 Na resonance fluorescence lidars at Beijing (40.2oN, 116.2oE), Hefei (31.8oN, 117.3oE), Wuhan (30.5oN, 114.4oE), and Haikou (19.5oN, 109.1oE). I know of two stations near Beijing: Yanqing (40.5°N, 116.0°E) and Pingquan (41.0°N, 118.7°E). Is the Beijing (40.2oN, 116.2oE) different from these two or it includes both? Please clarify.

Authors' response: The Na resonance fluorescence Beijing lidar is located at Yanqing. Different versions of the location of the station exist in publications. We double checked the location, and it is (40.5°N, 116.0°E). Therefore, the figures 10, 11, 13, and 15 are revised as the measurements of Na layers from OSIRIS made within ± 5∘ latitude and longitude square centred on Beijing station are changed.

Changes: Figures 10, 11, 13, and 15.

Other points:
Line 87: "the long-term routine measurements" is miss-leading as the word "long-term" has other connotations, as in "long-term linear trend". I would say "multi-year routine measurements", "continuing routine measurements", or simply "routine measurements".
Authors' response: corrected.
Changes: Line 91.

Line 122: Please define "S4 index".
Authors' response: done.
Changes: Line 129.

Line 171: "104oW" should be "105oW".
Authors' response: corrected.
Changes: Line 184.

Line 177: What is "ares"?
Authors' response: corrected to "areas".
Changes: Line 190.

Line 186: Please explain what is "and Na reference by Plane (2010)."?
Authors' response: done.
Changes: Line 200.

Line 191: Change "Na number density" in "Figure 9c shows the Na number density" by "Na density profile" or "the height dependent Na density".
Authors' response: done.
Changes: Line 205.

Line 204: Should "major reservoir" be "major reservoir of the neutrals"?
Authors' response: Yes, it has been revised.

Changes: Line 217.

Line 254: By "analyzed later", do you mean "analyzed below"?
Authors' response: analyzed in Discussion section.
Changes: Line 270.

Line 259: What happens to Fig. 11(g)? It appears empty in the figure.
Authors' response: No OSIRIS data at Haikou location. It has been marked in Fig. 11(g) in the revision.
Changes: Figure 11(g).

Line 286: In figure 14, do we compare the data under sunlit condition (OSIRIS) to nocturnal observations (Lidar) and full diurnal means (WACCM), correct? Would this be problematic?
Authors' response: In the revision, we compare the data from OSIRIS at ~6 LT, Na lidar at 4-6 LT and WACCM-Na at 6 LT. The comparison at 18 LT is not shown because the OSIRIS measurements at 18 LT are much less than those at 6 LT.
Changes: Figure 13 (in the revised manuscript).

Line 316: Do you have an explanation for why "an annual change in the Na column number density, with a summer minimum and a winter maximum"?
Authors' response: It has been explained in Figure 5 in 3.1 Global map of Na layer. "The significant summer-time depletion of high-latitude Na layers is primarily attributed to the temperature dependence of neutral Na chemistry with the very low temperature at the summer mesopause because of dramatic adiabatic cooling of upwelling air (Plane, 2003) and secondarily attributed to efficient removal of metallic species on noctilucent cloud particles (Plane et al., 2004; Raizada et al., 2007; Plane, 2012)". This is also our content of next research to investigate in detail, combined with the satellite and ground-based observations of Na and Es and WACCM simulations.

Line 318: Should you delete "Hefei," at the end?
Authors' response: It has been corrected by replacing "Hefei" with "Haikou".
Changes: Line 337.

Line 321: Should "lidaras" be "lidars".
Authors' response: corrected.
Changes: Line 339.

Line 334: Should "Na concentration" be "Na column density"?
Authors' response: corrected.
Changes: Line 358.